# How Far Are We from Intelligent Visual Deductive Reasoning?

**Yizhe Zhang**,[*]   **He Bai**,[*]   **Ruixiang Zhang**,[*]   **Jiatao Gu**,
**Shuangfei Zhai**,   **Josh Susskind**,   **Navdeep Jaitly**
Apple
{yizzhang,hbai7,ruixiangz,jgu32,szhai,jsusskind,ndjaitly}@apple.com

## Abstract

Vision-Language Models (VLMs) have recently demonstrated incredible strides on diverse vision language tasks. We dig into vision-based deductive reasoning, a more sophisticated but less explored realm, and find previously unexposed **blindspots** in the current SOTA VLMs. Specifically, we leverage Raven's Progressive Matrices (RPMs), to assess VLMs' abilities to perform multi-hop relational and deductive reasoning relying solely on visual clues. We perform comprehensive evaluations of several popular VLMs employing standard strategies such as in-context learning, self-consistency, and Chain-of-thoughts (CoT) on three diverse datasets, including the Mensa IQ test, IntelligenceTest, and RAVEN. The results reveal that despite the impressive capabilities of LLMs in text-based reasoning, we are still far from achieving comparable proficiency in visual deductive reasoning. We found that certain standard strategies that are effective when applied to LLMs do not seamlessly translate to the challenges presented by visual reasoning tasks. A detailed analysis reveals that VLMs struggle to solve these tasks mainly because they are unable to perceive and comprehend multiple, confounding abstract patterns in RPM examples.

## 1 Introduction

Recent advancements in Vision-Language Models (VLMs) have showcased the success of models such as GPT4-V (OpenAI, 2023) and Gemini (Team et al., 2023) across various vision language tasks. These tasks include captioning, object localization, multimodal world knowledge and commonsense, visual question answering (VQA), and vision-based coding (Yang et al., 2023). Previous evaluations of these models have proven that state-of-the-art (SOTA) VLMs are capable of performing well in numerous vision-based reasoning and understanding tasks (OpenAI, 2023; Team et al., 2023). Notably, prior works have demonstrated that strong VLMs can accurately extract text from images, understand and reason with charts and tables, and solve simple visual math problems (Yang et al., 2023; Nahida Akter et al., 2023).

In this study, we aim to evaluate the limitations of VLMs on challenging tasks that demand sophisticated vision-based deduction abilities, an area that has been relatively unexplored. Specifically, we ask the models to complete a set of Raven's Progressive Matrices (RPMs) problems (Kunda et al., 2013; Zhang et al., 2019), which are frequently used to measure human intelligence, by identifying the correct pattern to fill in the blank from multiple options. See Figure 1 for illustration. This requires the models to 1) comprehend each given pattern including the choices, 2) deduce underlying rules and identify any trend that can explain the evolution of these patterns, and 3) employ the learned rules to choose the missing pattern from the given options. The model's capacity to handle each aspect must be effectively coordinated to provide the correct answer. Our findings reveal that although some problems may seem intuitive to humans, they might not be as intuitive to VLMs.

---

[*] Equal contribution.

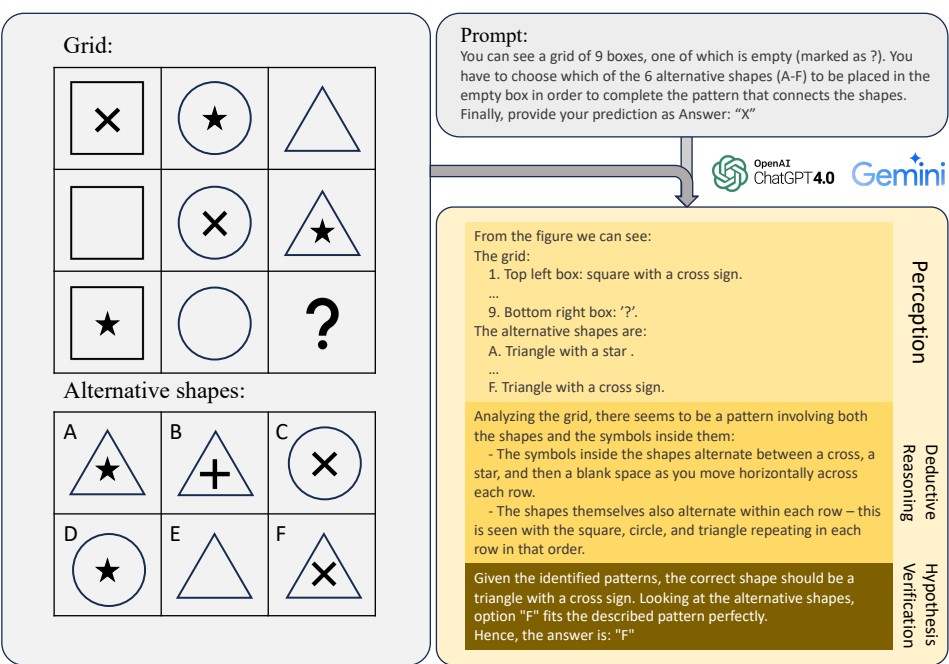

Figure 1: Illustration of the visual deductive reasoning for Raven's Progressive Matrices. The task requires intricate coordination among perception, deductive reasoning, and hypothesis verification capabilities exhibited by Vision-Language Models.

Compared to standard image reasoning tasks like VQA (Antol et al., 2015), RPMs pose several unique challenges: 1) RPMs require sophisticated deductive capabilities that involve multi-hop comparative reasoning, such as discrimination, relation, and analogy, while VQA typically requires only few steps of reasoning, 2) RPMs rely solely on visual clues to generate hypotheses and verify them, while VQA often involves using natural language to infer the objective and determine which parts to focus on, 3) RPMs are inherently few-shot (mostly 2-shot) learning tasks. Each RPM problem may have different underlying rules, which demands strong generalization abilities to solve them. Humans have a remarkable ability to learn from just a few examples, and powerful language models like LLMs have demonstrated this ability in text-based tasks. However, the ability of strong VLMs to solve few-shot reasoning tasks by relying solely on visual cues has not been well studied.

As an emerging field, it is crucial to establish benchmarks and systematic evaluations in order to push the limits of the visual deductive ability of VLMs. Our contributions include:

- We set up a framework for systematically evaluating VLMs on RPM problems. We evaluated several SOTA open-source and closed-source VLMs on three diverse datasets, including the Mensa IQ test, IntelligenceTest, and RAVEN, providing a comprehensive assessment of their performance. The results indicate that although LLMs exhibit impressive capabilities in text-based reasoning, such proficiency has not been achieved in image-based reasoning. The code and evaluation datasets have been released to facilitate future investigation and improvement over VLMs.
- We employed standard inference-time strategies in LLMs such as in-context learning (Brown et al., 2020) and self-consistency (Wang et al., 2022) to probe the potential of VLMs. We found that some standard strategies that are effective in LLMs do not seamlessly translate to the VLMs we used.
- We finely diagnose the performance bottleneck of VLMs by breaking down their capability into *perception*, *deductive reasoning*, and *hypothesis verification*. Our analysis reveals that perception is the limiting factor in current VLMs. To scrutinize this specific "blind spot" in strong VLMs such as GPT-4V, we provide a case study highlighting where issues occur.

- We identified and examined several issues associated with the current VLMs in this task. These issues include overconfidence, sensitivity to prompt design and an inability to effectively leverage in-context examples. We ablated the effects of different prompts on the overall performance of the model and found models can benefit from more *structured* prompts.

## 2 Related Work

**General LLM Reasoning benchmarks** Many text-based reasoning tasks and benchmarks have been introduced to evaluate LLMs in various domains (Huang & Chang, 2022) such as general knowledge (Hendrycks et al., 2020), math reasoning (Cobbe et al., 2021), common-sense reasoning (Geva et al., 2021; Clark et al., 2018), factual reasoning (Laban et al., 2023), and coding (Chen et al., 2021). Some noteworthy examples of these works are BIG-bench (Srivastava et al., 2022), HELM (Liang et al., 2022) and SuperGLUE (Sarlin et al., 2020).

**Visual reasoning evaluation** Previous work on visual reasoning tasks has primarily focused on tasks such as visual question answering (VQA) Antol et al. (2015) and image captioning. These tasks involve answering questions about images or generating natural language descriptions of visual content. Researchers have also examined the ability of models to understand the relational and compositional aspects of objects in images. Datasets like CLEVR (Johnson et al., 2017) and SHAPES (Andreas et al., 2016) assess visual reasoning abilities such as counting, comparing, logical reasoning, and storing information in memory. As the VLMs abilities to perform visual reasoning have evolved so have the benchmarks. New benchmarks, like MMMU (Yue et al., 2023) and MathVista (Lu et al., 2023) have been developed that test the models' ability to emulate human-like understanding of scenes and objects in images and videos. These benchmarks include areas such as scene text understanding (Sidorov et al., 2020; Schiappa et al., 2024), formulation (**?**), table and chart interpretation (**?**), the comprehension of visual stimuli (Yang et al., 2023), geometric reasoning (Ahrabian et al., 2024), spatial reasoning (Chen et al., 2024), and facial expression comprehension and reasoning (Yang et al., 2023).

This paper focuses on RPMs. Our goal was to simulate a more holistic challenge that mirrors scenarios where generalist VLMs must navigate through even unseen and unfamiliar scenarios. Although the RPMs were designed for humans, they represent a fundamental type of visual-spatial reasoning that artificial intelligence systems, particularly those aimed at achieving general intelligence, should also be able to perform well on.

**Deductive reasoning** Deductive reasoning evaluation and benchmarks have been conducted for both textual and visual domains. Two notable examples are GuessWhat?! (De Vries et al., 2017) and ReferIt (Kazemzadeh et al., 2014), which assess the visual reasoning abilities of the models being tested. More recently, LMRL Gym (Abdulhai et al., 2023) and Entity Deduction Arena (Zhang et al., 2023) have been introduced as methods to evaluate the ability of LLMs to perform multi-turn deductive reasoning tasks. Another relevant task is ARC (Acquaviva et al., 2022) which shares similarities with RPMs, as they both require correctly inferring unseen outputs based on given examples. Comparing with ARC, RPMs are abstract and requires intricate analogical and relational reasoning.

Using RPMs to assess VLM is not rare. For instance, many general VLM benchmarks like MATHVISTA (Lu et al., 2023) and MMMU (Yue et al., 2023) uses RPMs problems to probe models' reasoning and understanding ability of complex patterns. When GPT-4v came out, technical evaluation work (Yang et al., 2023) also probed its performance on abstract patterns, especially the RPMs. Different from them, our work dives deep into scrutinizing the blindspot, and the underlying issues when VLMs deals with these abstract patterns, showing some potential directions for future improvement.

## 3   Experiment Setting

**Dataset** In our paper, we employed three RPMs datasets. The **Mensa Test**[1] consists of 35 questions with progressive levels of difficulty. For the purpose of 1-shot learning, we used the first question as an in-context example and reserved the remaining 34 questions for evaluation. The **IntelligenceTest** (IT)[2] provided an IQ test encompassing verbal, pattern recognition, math, and structural components. We specifically focused on pattern recognition, which solely comprised RPMs problems and included 66 examples. Additionally, we incorporated the **RAVEN** dataset (Zhang et al., 2019) for evaluation. The RAVEN dataset employs a generative model to create RPMs problems using a hierarchical pipeline. The test dataset of RAVEN contains 14,000 examples, covering 7 types of distinct figural configurations that incorporate different layouts, shapes, and relational structures. In this work, we generate 140 new samples, 20 samples for each figural configuration.

**Models** We compared various VLMs that represent the state-of-the-art for both closed-source and open-source models, including GPT4-V (gpt-4-vision-preview) (OpenAI, 2023), Gemini-pro (Team et al., 2023), Qwen-VL-Max (Bai et al., 2023), LLaVA-1.5-13B and LLaVA-1.6-34B(Liu et al., 2023). We use the default sampling method for each of the tested VLMs in our generation process, show in Appendix A.3.

**Prompts** We prompt the model with the instruction followed by the query image. We provide the prompt in the Appendix A.2.

## 4   Evaluation Results

### 4.1   Evaluation of VLMs on Visual Deductive Reasoning

| | Mensa | | IntelligenceTest (IT) | | RAVEN | |
|---|---|---|---|---|---|---|
| | **Entropy** | **Accuracy↑** | **Entropy** | **Accuracy↑** | **Entropy** | **Accuracy↑** |
| GPT-4V | 1.49 | $0.24 \pm 0.05$ | 1.40 | $0.16 \pm 0.04$ | 2.07 | $0.12 \pm 0.04$ |
| Gemini Pro | 1.24 | $0.15 \pm 0.04$ | 1.18 | $0.18 \pm 0.03$ | 1.37 | $0.11 \pm 0.04$ |
| QWen-VL-Max | 1.13 | $0.17 \pm 0.01$ | 0.97 | $0.13 \pm 0.02$ | 0.48 | $0.10 \pm 0.03$ |
| LLaVA-1.5-13B | 0.72 | $0.23 \pm 0.01$ | 0.64 | $0.09 \pm 0.01$ | 0.25 | $0.10 \pm 0.03$ |
| LLaVA-1.6-34B | 0.81 | $0.22 \pm 0.01$ | 0.78 | $0.11 \pm 0.01$ | 0.25 | $0.10 \pm 0.03$ |
| Random Guess | 2.58 | 0.16 | 2.58 | 0.16 | 3.00 | 0.12 |

Table 1: Benchmark of VLMs on three different datasets. "Entropy" denotes uncertainty of the prediction, and "Accuracy" indicates the percentage of accurately answered questions.

In Table 1 we show how different VLMs performed on each dataset. For each model and dataset, we computed the statistics by averaging them over 10 repetitions. From the table, it is evident that GPT-4 either slightly surpasses or is on par with the other models across all benchmarks. However, the accuracy gap between the models is not substantial in terms of their ability to solve RPM puzzles. It is interesting to note that the performance of these models is comparable to random guessing (last row), indicating their limited effectiveness in this area. Converting the accuracy on the questions to human ranking scale, we find that the models rank in the 2-8 percentile on the Mensa tests. On the IT dataset humans demonstrate a wide range of success rates per question, spanning from 30% to 93.4%, which is much higher than the highest accuracy of a mere 18% observed for Gemini Pro. Similarly, on the Raven dataset humans attain an impressive success rate of 84.67% (Zhang et al., 2019), starkly outperforming VLMs, which consistently yield results akin to random guessing.

**Uncertainty of the prediction** We analyze the entropy of model predictions in order to assess the uncertainty inherent in their predictive distribution. For the choices set $C$, the

---

[1]https://www.mensa.org/public/mensa-iq-challenge
[2]https://www.intelligencetest.com/questions

Entropy is defined as $S = -\sum_{i \in C} p_i \log p_i$. If the model consistently predicts a single answer, it is has an entropy of 0. If it randomly guesses, the entropy reaches the upper bound shown in the Table 1. For Self-consistency model, we bootstrapped 1000 leave-one-out repetitions from 5 answers per question and calculated entropy over aggregated predictions from each repetition.

We see that GPT-4 and Gemini Pro exhibit a greater diversity of answers, which is also reflected in the greater diversity in recognizing and attempting to identify various patterns. On the other hand, LLaVA and QWen-VL produce more deterministic predictions, resulting in lower entropy.

Interestingly, we observed that even when the entropy was high, models tried to provide a nonsensical rationale, instead of acknowledging their inability to perform the task; this was observed to happen more often with models that had higher entropy. All the tested models never express any level of uncertainty by using words like "likely" or "maybe". This excessive confidence can presumably be attributed to the model pretraining and instruction finetuning steps, which typically do not involve calibrating the model for uncertainty. Instead, the models are encouraged to generate uncertain content, leading to more errors in aggregating in the generated output.

## 4.2 Do standard strategies in LLMs translate effectively to visual deductive reasoning?

We tried two strategies effective in LLMs: 1) **1-shot** (Brown et al., 2020) prompts in-context RPM example and its solution to the VLMs. 2) **Self-consistency** (SC) (Wang et al., 2022) samples multiple responses and selecting the majority voted answer.

| | **Mensa** | | **IntelligenceTest** | | **RAVEN** | |
|---|---|---|---|---|---|---|
| | **Entropy** | **Accuracy↑** | **Entropy** | **Accuracy↑** | **Entropy** | **Accuracy↑** |
| GPT-4V (0-shot) | 1.49 | $0.24 \pm 0.05$ | 1.40 | $0.16 \pm 0.04$ | 2.07 | $0.12 \pm 0.04$ |
| GPT-4V (1-shot) | 1.41 | $0.22 \pm 0.06$ | 1.31 | $0.17 \pm 0.04$ | 2.03 | $0.12 \pm 0.04$ |
| GPT-4V (SC) | 0.17 | $0.31 \pm 0.01$ | 0.15 | $0.19 \pm 0.02$ | 0.20 | $0.10 \pm 0.02$ |
| Gemini Pro (0-shot) | 1.24 | $0.15 \pm 0.04$ | 1.18 | $0.18 \pm 0.03$ | 1.37 | $0.11 \pm 0.04$ |
| Gemini Pro (1-shot) | 0.69 | $0.17 \pm 0.03$ | 0.54 | $0.19 \pm 0.01$ | 1.35 | $0.10 \pm 0.03$ |
| Gemini Pro (SC) | 0.03 | $0.18 \pm 0.01$ | 0.03 | $0.18 \pm 0.01$ | 0.08 | $0.10 \pm 0.01$ |

Table 2: Expanded benchmark of VLMs on three different datasets, including the 1-shot and SC variants for both GPT-4 and Gemini models. The prompts are provided in Appendix A.2.

**VLMs struggle with reading in-context image** The performance of the 1-shot evaluation, shown in Table 2, did not demonstrate improvement compared to the 0-shot evaluation. Specifically, we observed only a marginal 1% enhancement for the IntelligenceTest dataset, while encountering a decrease of 2-4% in accuracy for the Mensa test. Surprisingly, all the tested models, including GPT-4V and Gemini, struggle with a high failure rate *even when the in-context example is identical to the current task being solved*. This is peculiar because powerful LLMs usually exhibit

| In-context | Query | Accuracy |
|---|---|---|
| Desc. + Rat. + Ans. | Desc. | 100% |
| Img. + Desc. + Rat. + Ans. | Img. + Desc. | 80% |
| Img. + Desc. + Rat. + Ans. | Img. | 20% |
| Img. + Ans. | Img. + Desc. | 80% |
| Img. + Ans. | Img. | 40% |

Table 3: GPT-4V analogizes better when solely based on text descriptions. Desc., Rat., Ans. and Img. represents description, rationale, answer and image, respectively

the ability to analogize and copy the in-context example when provided with the same query. We observed accuracy ranging from 10% to 20% for these in-context examples across different datasets, which is comparable to the accuracy when a different example is used as the in-context example.

In order to make this observation concrete we present an ablation experiment with a specific example we created manually in the style of Mensa problems, which we call *M-easy* (See Figure 2a for the problem and Table 3 for a summary of results). Here the same example is used as the in-context example, and as the task being solved, the model only needs to be

able to draw a comparison between the in-context example and the query, and copy over the answer from the in-context sample[3].

We first cast the problem as a text-only problem using appropriate descriptions for both the in-context example and the query (row 1). The model demonstrates a perfect accuracy of 100% showing that it is easy for it to solve this problem when it is represented as text. Next, we added the image to the textual description for both the in-context example and the query. The accuracy now decreases to 80%, even though additional visual information has been provided (row 2). Finally, when the text description is removed from the query, the accuracy significantly drops to 20% (row 3). We hypothesize that the drop in accuracy arises because it is much harder for the model to compare image tokens than it is to compare textual tokens and also that the model utilizes text more than it does the images.

## 5   What limits the performance of the VLMs?

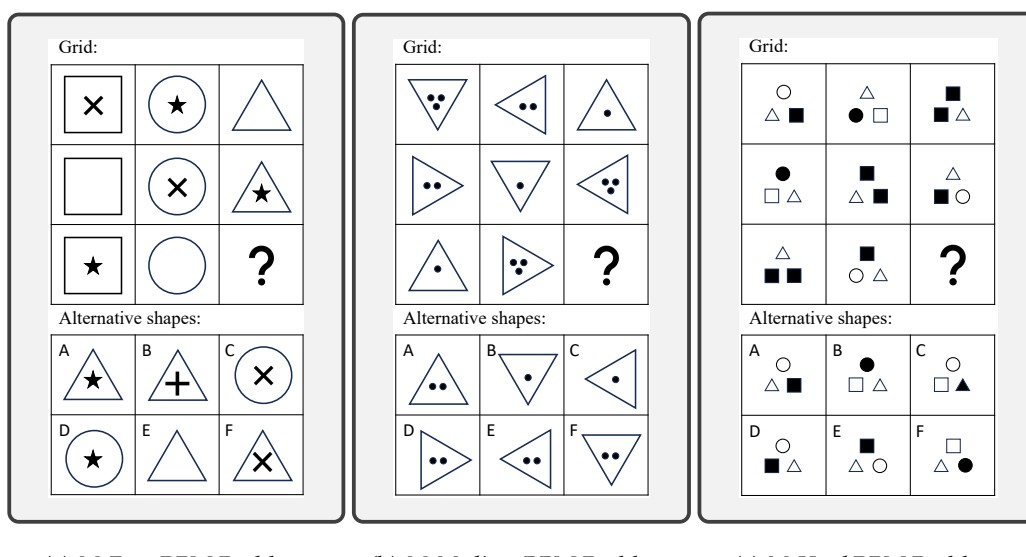

(a) *M-Easy* RPM Problem    (b) *M-Medium* RPM Problem    (c) *M-Hard* RPM Problem

Figure 2: Three manually created RPM problems evaluated for text description augmentation, illustrating varying levels of difficulty. The correct answers are "F, F, F".

We investigate why VLMs fail to reach human-level performance in answering even simple questions that are intuitive to humans. For this purpose, as a case study, we manually created three RPMs with varying degrees of difficulty, as depicted in Figure 2. The manually curated examples are similar to instances in the larger datasets (e.g. Mensa), whose example cannot be presented due to copyright issues. We aim to use these examples as qualitative probes to showcase the blind spots in depth, and show that the issues are ubiquitous in VLMs, spanning stages of perception, reasoning, and verification. To conduct a fine-grained analysis and diagnosis of the VLM's inability to perform this task of visual deductive reasoning with RPMs, we decompose the evaluation into three consecutive stages: 1) **Perception**: assess if the model can understand and describe the patterns in the RPMs; 2) **Deductive reasoning**: evaluate if the model can discern and articulate underlying rules; 3) **Hypothesis verification**: examine the model's proficiency in formulating a plausible hypothesis for the missing pattern and identifying a matching option among alternatives.

### 5.1   How good is the VLM's perception on this task?

We first asked the model to describe the RPM figures, to assess if they understood the images that were provided as part of the problem. Surprisingly, even though VLMs are astoundingly

---

[3]The results are based on 10 repetitions

| Position | Description of *M-Easy* RPM | Description of *M-Medium* RPM | Desc. of segmented *M-Medium* RPM |
|---|---|---|---|
| Top left | A square with an X inside. | Triangle pointing down with three dots forming a vertical line in the center. | Inverted triangle with three dots inside. |
| Top center | A circle with a star inside. | Triangle pointing right with three dots forming a horizontal line along the center. | Right-pointing triangle with two dots. |
| Top right | An empty triangle. | Triangle pointing up with four dots forming a vertical line in the center. | Upright triangle with one dot in the center. |
| Middle left | A square with an X inside. | Triangle pointing down with two dots forming a horizontal line in the middle. | Right-pointing triangle with two dots. |
| Middle center | A circle with an X inside. | Triangle pointing right with a single dot in the center. | Inverted triangle with one dot in the center. |
| Middle right | A triangle with an X inside. | Triangle pointing up with two dots forming a vertical line along the center. | Left-pointing triangle with three dots. |
| Bottom left | A square with a star inside. | Triangle pointing down with one dot in the center. | Upright triangle with one dot in the center. |
| Bottom center | A circle. | Triangle pointing right with two dots forming a horizontal line in the middle. | Right-pointing triangle with three dots. |

Table 4: The M-Easy and M-Medium RPMs descriptions from GPT-4V for the patterns can contain errors, including hallucinations and Chimera descriptions. When the model is provided with segmented RPM images (i.e., when patterns are separated into multiple image inputs), it leads to a reduction in the error. Errors are indicated in red.

accurate in describing commonplace images, they seemed to be quite unsuccessful at accurately describing even the simpler abstract patterns we gave them. The generated descriptions contained numerous errors *across all the tested models*, as exemplified by results from GPT-4V in Table 4. More examples are shown in Appendix A.5. We identified two major issues for this *blind spot* of VLMs:

**Compounding error**: Models tend to replicate the descriptions of previous patterns, leading to an autoregressive amplification of compounding errors in successive descriptions. This results in an increasingly erroneous narrative throughout the generation process. For example, in Table 4 (M-Medium), When the model first makes a mistake by including "a vertical line" in the description, the subsequent text follows the same error. We think that the autoregressive nature of the VLMs causes it to repeat itself, with the preceding text dictating the entire follow-up text.

**Confounding error**: The similarities between patterns cause confusion, as the model struggles to maintain focus on a single pattern. Consequently, we often observe "Chimera descriptions" that erroneously combine elements from multiple patterns. For example, in Table 4 (M-Easy, middle right), the description seems to combine elements in two adjacent patterns (middle center, middle right). This could be attributed to the model's failure to effectively focus its attention on the corresponding pattern when all the patterns appear similar.

These two issues are prevalent across all the methods and datasets. When the patterns contain multiple elements and are more detailed, these issues become severer.

**Can decomposing the RPMs into each single pattern from the grid enhance perception?** Presumably, by decomposing the patterns into individual components, we can eliminate the *confounding* errors. To investigate this, we first segmented each of the three manual examples shown in Figure 2, into 9 individual question patterns and 6 candidate patterns. We then used a new prompt A.2 for

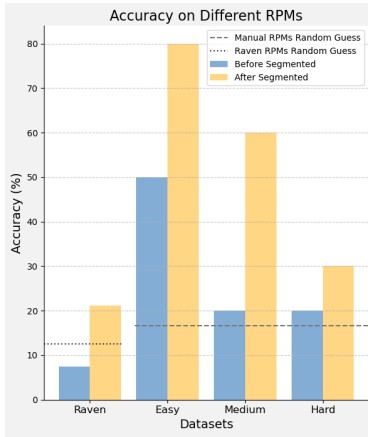

Figure 3: Accuracy of the original RPM as input with that of the segmented RPM as input. Results based on 10 repetitions.

GPT-4V to read both the full image and the segmented patterns to infer the answer. In this way, we found GPT-4V would describe each pattern more accurately. The descriptions of the *M-Medium* RPM can be found in Table 4. We conducted 10 tests for each RPM and report the accuracy comparison with and without segmentation in Figure 3. We also verify the segmentation impact using the Raven dataset (60 examples). We got 21.2% accuracy for segmented RPMs and 7.4% for non-segmented RPMs. The results demonstrate a significant reduction in *confounding* errors, confirming the issues discussed earlier.

**Hallucination** We have also observed the model generating hallucinations in the descriptions, particularly regarding counting. For instance, in Table 4 (M-Medium, top right), the model erroneously states that there are four dots when, in reality, there are only three.

**Data distribution aspect** VLMs are presumably trained primarily on naturalist images, which may cause them to be less sensitive to abstract patterns. However, evaluation tasks on foundation models often involve significant distribution shifts or completely novel tasks, and foundational models like VLMs are expected to have some capacity for task generalization and zero-shot capabilities. RPM puzzles challenge VLMs to engage in abstract reasoning and pattern recognition, serving as a controlled test for perception, self-reflection, and deductive reasoning abilities. Additionally, IQ evaluations for humans already use abstract visual puzzles, establishing a comparative connection to human performance.

While we believe that additional finetuning could potentially improve the performance, we hypothesize that finetuning the model with RPMs might not entirely eliminate the *compounding* and *confounding errors*, as they appear to be inherent limitations of the VLMs from training.

## 5.2 How good is the VLM's deductive reasoning on this task?

Next, we assess the model's ability to perform effective reasoning by conditioning it on the ground truth text description of the RPMs. We provide the prompts in Appendix A.2.

**Does the oracle text description improve the model's performance?** The original evaluation (Tables 1 and 2) requires the model to directly generate the answer, making it difficult to disentangle the understanding and deductive reasoning aspects. To examine the VLMs more closely, we provided each evaluated model with oracle text descriptions that were manually created by the authors. We then evaluated the models' performance on the three RPM problems and present the results in Table 5 (GPT-4V + Oracle Desc.). The oracle text descriptions can be found in the Appendix A.4. We also provide sampled rationale generated by GPT-4V in the Appendix A.6.

It is evident that the model's performance has been significantly improved with the addition of oracle descriptions for each pattern (Table 5). The models are able to analyze the given patterns and deduce rules for the *M-Easy* and *M-Medium* RPMs, and provide rationale for the problem. For the *M-Hard* RPM, the models demonstrate some capability of reasoning, albeit with some challenges and is far from human parity. We provide additional examples in the Appendix. However, it is not clear whether the models still rely heavily on visual cues or if their reasoning is purely text-based.

**Will removing the visual cues harm the model?** Next, we examine whether textual information alone is sufficient by removing the visual information. The results, shown in Table 5 (GPT-4V + Oracle Desc. - Visual), are intriguing. Without visual information, the models can maintain a similar level of performance for *M-Easy* and *M-Medium* RPMs. Notably the result solely rely on the textual information of the input is superior to the GPT-4V baseline, which mostly rely on visual information of the input. However, as the tasks become more challenging (*M-Hard* RPM), the models start to struggle. The performance is also worse than GPT-4V baseline. This suggests that for tasks that involve complex spatial layouts and relational reasoning, text alone may be insufficient and potentially confusing, while visual cues may provide additional visual *alignment* and better *comparative* attention. In such cases, visual information and textual clues would complement each other and work in synergy to achieve the optimal performance. Interestingly, when we

| Model | Acc. | Ent. | A | B | C | D | E | F |
|---|---|---|---|---|---|---|---|---|
| *M-Easy* | | | | | | | | |
| GPT-4V 1-shot | 50% | 1.69 | 0 | 0 | 1 | 1 | 3 | 5 |
| GPT-4V 1-shot + Gen. Desc. (CoT) | 50% | 1.36 | 0 | 1 | 0 | 0 | 4 | 5 |
| GPT-4V 1-shot + Oracle Desc. | 60% | 1.57 | 0 | 1 | 0 | 1 | 2 | 6 |
| GPT-4V 1-shot + Oracle Desc. - Visual | 60% | **0.97** | 0 | 4 | 0 | 0 | 0 | 6 |
| GPT-4V 1-shot + Oracle Desc. + Rationale | **60%** | 1.57 | 1 | 0 | 0 | 1 | 2 | 6 |
| *M-Medium* | | | | | | | | |
| GPT-4V 1-shot | 20% | 2.25 | 0 | 3 | 2 | 2 | 1 | 2 |
| GPT-4V 1-shot + Gen. Desc. (CoT) | 50% | 1.69 | 0 | 3 | 1 | 1 | 0 | 5 |
| GPT-4V 1-shot + Oracle Desc. | **80%** | 0.92 | 1 | 1 | 0 | 0 | 0 | 8 |
| GPT-4V 1-shot + Oracle Desc. - Visual | 60% | 1.57 | 1 | 0 | 0 | 1 | 2 | 6 |
| GPT-4V 1-shot + Oracle Desc. + Rationale | 70% | **0.88** | 3 | 0 | 0 | 0 | 0 | 7 |
| *M-Hard* | | | | | | | | |
| GPT-4V 1-shot | 20% | 1.49 | 0 | 0 | 3 | 0 | 5 | 2 |
| GPT-4V 1-shot + Gen. Desc. (CoT) | 30% | 1.97 | 0 | 2 | 2 | 0 | 3 | 3 |
| GPT-4V 1-shot + Oracle Desc. | 40% | 1.85 | 0 | 3 | 0 | 1 | 2 | 4 |
| GPT-4V 1-shot + Oracle Desc. - Visual | 10% | 1.96 | 2 | 0 | 1 | 5 | 1 | 1 |
| GPT-4V 1-shot + Oracle Desc. + Rationale | **50%** | **1.49** | 0 | 2 | 0 | 0 | 3 | 5 |

Table 5: Breakdown of GPT-4V variants with augmented text description across different RPMs. Each combination is ran for 10 repetitions. The correct answer "F" is marked in color. Given budget constraint we only test on the 3 manual examples for this analysis.

provide GPT-4V with an incorrect description, there is around an 80% chance that the model recognizes the mismatch between the text and the image and responds as: "There has been a misinterpretation of the provided image". The model, nevertheless, still generates some rationale which seems adhere more closely to the text description than to the visual cues.

**Can the performance be improved by reasoning with noisy text descriptions generated by the model itself?** Drawing inspiration from Chain-of-Thoughts (CoT) in the text domain (Wei et al., 2022) and the recent Self-Imagine work (Akter et al., 2024), we further investigate whether VLMs can enhance their performance using noisy text descriptions that they generate on their own. This also helps us understand the extent to which VLM reasoning relies on accurate descriptions of images and the extent to which it can recover from errors in the descriptions. Table 5 (GPT-4V + Gen Desc.) shows that incorrect text descriptions can still produce a gain. The gap between self-generated descriptions and oracle descriptions, however, varies across the different cases.

## 5.3 How good is the VLM's hypothesis verification on this task?

Finally, We tested the performance of GPT-4V when it received both an oracle description and an oracle rationale. The oracle rationale, which can be found in Appendix A.2, only includes the explanation of the underlying rule without predicting the final pattern or answer. The results for 10 repetitions on manual examples are shown in Table 5 (GPT-4V + Oracle Desc. + Rationale). Surprisingly, compared to the row representing GPT-4V + Oracle Desc., the oracle rationale did not significantly improve accuracy. In cases where the model failed, it sometimes directly generated an incorrect answer and at other times extended the rationale but still generated false answers. For example, for M-easy, GPT-4V continued to generate "the third row should have a star, as the first two boxes of the third row (square and circle) already have a star." This indicates that hypothesis generation and verification are closely tied to deductive reasoning, and the model has not yet reached human-level performance in following hints and turning learned rules into future predictions.

Interestingly, strong models like GPT-4V exhibit some strategies similar to humans. For instance, they often use the answer options along with the grid to form and tests hypotheses, rather than generating a hypothesis solely based on the grid and then checking for

any matches with the alternative shapes.[4] GPT-4V also sometimes employs a strategy of elimination to rule out incorrect answers (e.g., "the right shape should have a cross sign, which leaves the options to C and F.").

### 5.4 How does the prompt format influence the model prediction?

The format of the prompt can sometimes significantly impact the performance of VLM. For example, we found the arrangement of task instruction and images is crucial to Gemini Pro. We show the results in Table 6. We observed a remarkable 200% increase in prediction accuracy when we simply altered the sequence of these elements. However, we don't observe similar conclusion from other tested models.

| Prompting Structure | Mensa |
|---|---|
| Gemini Pro *Image First* | $2.3 \pm 1.3$ |
| Gemini Pro *Instruction First* | $5.4 \pm 1.2$ |
| GPT4V 1-Shot *w/o Sentinel Token* | $6.1 \pm 1.5$ |
| GPT4V 1-Shot *w/ Sentinel Token* | $7.8 \pm 1.7$ |

Table 6: Average number of correct predictions made by GPT4-V and Gemini Pro on the Mensa test, demonstrating its sensitivity to the structure of prompts used.

We also delves into the differences in how the model performs under 0-shot and 1-shot evaluation setups. We discovered that using special sentinel tokens, such as [{BEGIN/END}_OF_EXAMPLE] to separate text prompts from images helps the model delineate task instructions from in-context examples. This method of structuring prompts is particularly effective in aiding the model's comprehension across all tested VLMs. For instance, we show the results of GPT-4V in Table 6. Experiment results of k-shot evaluations are detailed in Appendix A.1.

This study underscores that VLMs, unlike their text-only counterparts, can benefit from a more *structured* format in their task prompts. Furthermore, the interaction between different modalities, such as text and image, needs to be carefully considered and evaluated.

## 6 Conclusion

This work is a systematic evaluation of the performance of popular Vision-Language Models (VLMs) in a variety of Raven's Progressive Matrices (RPMs). These tasks serve as a challenging benchmark for assessing the models' ability to reason based on visual clues. We observed that the current state-of-the-art VLMs still fall short of achieving human-level performance on these tasks, with the best-performing models being close-sourced. Our analysis of the models' performance reveals that perceptual understanding may be the main bottleneck, as the models perform better when provided with appropriate textual descriptions. In future work, it would be intriguing to validate our hypothesis concerning the blind spot of VLMs when it comes to describing patterns. This investigation has the potential to enhance the general recognition and attentiveness capabilities of VLMs. Additionally, exploring the development of contrastive learning or reinforcement learning algorithms could further improve the model's visual deductive reasoning abilities. To investigate whether the VLMs' struggle is due to a lack of general reasoning or data distributional shifts, future work can be done to evaluate VLMs on pattern-based reasoning tasks with naturalistic images, and investigate the impact of additional instruction fine-tuning on RPMs.

---

[4]This generate-then-verify strategy accounts for less than 10% of GPT-4V's behavior in our observation. In such cases the model often rejects the options provided and responds as follows: "Unfortunately, the given options do not correspond with the identified pattern."

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
