# OpenReview forum: "How Far Are We from Intelligent Visual Deductive Reasoning?"
_colmweb.org/COLM/2024/Conference — COLM_

### Official Review · Reviewer_1JTm · 2024-05-10

**Rating:** 7
**Confidence:** 4
**Ethics Flag:** 1

**Summary:**

This is an analytical study focusing on visual deductive reasoning. The authors utilized RPMs to assess the capabilities of VLMs in multi-hop relationships and deductive reasoning. They conducted comprehensive evaluations on several popular VLMs using standard strategies across three different datasets, resulting in some interesting conclusions.

**Questions To Authors:**

1. RPMs problems are similar to multiple-choice questions in NLP and represent some very basic inquiries. Perhaps additional questions beyond RPMs are needed to assess human intelligence or to evaluate VLMs.
2. Since the framework for systematically evaluating VLMs on RPM problems is treated as a contribution, there should be a clear explanation of this framework. However, I can't find any specific descriptions or relevant illustrations in this manuscript.

**Reasons To Accept:**

Analyzing visual deductive reasoning can be quite intriguing.

**Reasons To Reject:**

1. RPMs problems are similar to multiple-choice questions in NLP and represent some very basic inquiries. Perhaps additional questions beyond RPMs are needed to assess human intelligence or to evaluate VLMs.
2. Since the framework for systematically evaluating VLMs on RPM problems is treated as a contribution, there should be a clear explanation of this framework. However, I can't find any specific descriptions or relevant illustrations in this manuscript.

---

> ### Author Rebuttal · Authors · 2024-05-29
>
> We appreciate the thoughtful feedback and questions raised.
>
> **RPMs Similarity to Multiple-Choice Questions**: While RPMs may resemble multiple-choice questions, they represent a unique challenge in visual reasoning and abstract pattern recognition. RPMs are designed to measure abstract reasoning and pattern recognition, which are more complex than typical multiple-choice questions in NLP. RPMs require identifying relationships between visual elements and applying logical rules, making them a robust test for evaluating the reasoning capabilities of VLMs. However, we agree that expanding the evaluation to include other types of questions could provide a more comprehensive assessment of human intelligence and VLM performance. We will consider this in future work.
>
> **Explanation of the Framework**: We agree that a clearer explanation of our evaluation framework is warranted. The manuscript does contain a detailed explanation of our framework for evaluating VLMs on RPM problems (Figure 1). We will revise the manuscript to make this section more prominent and ensure that all relevant illustrations and descriptions are clearly presented to guide the reader through our systematic approach.

---

### Official Review · Reviewer_j4Q4 · 2024-05-12

**Rating:** 9
**Confidence:** 5
**Ethics Flag:** 1

**Summary:**

This paper probes visual deductive reasoning with Raven's progressive matrices, a widely used set of non-verbal intelligence tests, and shows that even the latest VLM's fail to do well on visual deductive reasoning. The authors perform thorough evaluations of a variety of large visual models and establish a clear gap in VLM performance on visual deductive reasoning. They go on to disentangle the performance into perception, deductive reasoning and hypothesis verification and find that the VLM's fall short on perception. The paper is well written and the insights are valuable to the community at large and are novel.

**Questions To Authors:**

Well written paper with interesting insights.

**Reasons To Accept:**

1. Well motivated paper that situates the proposed work well in the literature through a thorough literature survey.
2. Novel insights stemming from thorough and careful evaluation.
3. Interesting decomposition into perception, deductive reasoning and hypothesis verification.

**Reasons To Reject:**

These are minor weaknesses that can be easily addressed in the camera ready and therefore are not true reasons for rejection
1. The authors should look at the following work on natural images. The authors' proposed work with Raven's progressive matrices is quite different but some comments would be good
https://arxiv.org/abs/2304.03659
2. The authors' insights through the disentangling are consistent with the following (especially on GPT-4 V performance)
https://arxiv.org/abs/2304.03659
Some comments would be helpful

---

> ### Author Rebuttal · Authors · 2024-05-29
>
> Thank you for the feedback and the references provided. We appreciate the suggestions and will be happy to address them in the next version.
>
> **Additional References on Natural Images** we acknowledge the relevance of the work on natural images mentioned in the reference (https://arxiv.org/abs/2304.03659). While our work focuses specifically on abstract visual reasoning tasks like Raven's Progressive Matrices (RPMs), we agree that drawing connections or contrasting our findings with the broader domain of natural image understanding could provide valuable insights. We will include a discussion on the similarities and differences between our work and the referenced study, highlighting the unique challenges posed by RPMs and the potential applicability of our findings to other visual reasoning tasks.
>
> **Consistency with Disentangling Insights** we appreciate the suggestion to relate our insights to the findings reported in the reference, particularly concerning GPT-4V's performance. The cited work provides valuable insights into the general performance of GPT-4 V, but our focus is on a specific aspect of visual deductive reasoning that involves complex pattern recognition and multi-hop reasoning tasks. We will carefully review the relevant sections of the referenced work and incorporate a discussion on the consistency or divergence of our observations with their findings.

---

> > ### Comment · Reviewer_j4Q4 · 2024-06-06
> > **Response to rebuttal 1**
> >
> > I am satisfied with your rebuttal. I will maintain my original score. Please do follow through on the promises you have made in the rebuttal.

---

### Official Review · Reviewer_Hoxc · 2024-05-25

**Rating:** 5
**Confidence:** 4
**Ethics Flag:** 1

**Summary:**

This paper evaluates several VLMs on their ability to do visual deductive reasoning, by evaluating on Raven's Progressive Matrices (RPM) puzzles that involve reasoning about abstract visual patterns. They evaluate VLMs on three RPM datasets: MensaIQ, IntelligenceTest and RAVEN, and find that all VLMs perform poorly on these abstract visual reasoning problems. The authors conduct an in-depth analysis of why VLMs struggle with these tasks and find that the main bottleneck is perception

UPDATE 6/6/24: Updated overall score from 4 to 5.

**Questions To Authors:**

- how is the in-context example presented to models that cannot take interleaved visual inputs?
- is entropy calculated from the logits, or from multiple samples? It’s very unclear what p_i is in Section 4.1
- how is entropy calculated in the context of self-consistency?
- Why do the problems created by the authors in Figure 2 exhibit varying degrees of difficulty for the models?

**Reasons To Accept:**

- The division of visual deductive reasoning into perception+deductive reasoning+hypothesis verification, and the systematic experiment setups to diagnose these individual capabilities to identify the bottlenecks for VLMs, is well designed.

**Reasons To Reject:**

- **Analysis done with only a few examples**: The analysis in Table 3 (second half of Section 4.2) and Section 5 are performed on only 1--3 examples, which makes it hard to take any of the findings at face value. The compounding error and confounding error effects in Table 4 are each observed only once. The analysis of deductive reasoning capabilities in Section 5.2 and hypothesis verification in Section 5.3 are also conducted only on the three curated examples; while annotating oracle descriptions may be expensive, it can surely be done for more than 3 examples. I do not find any of the conclusions from the sections mentioned above to be compelling, given the very minimal amount of evidence to back those conclusions.

- **Insufficiently motivated setting**: Why are abstractive visual puzzles a useful testbed to examine VLMs' deductive reasoning capabilities? As the authors note in Section 5.1, these models are primarily trained on naturalistic images, so while these tasks may be useful for diagnosing the reasoning capabilities of humans, it's hard to say they cannot do deductive reasoning if they fail to perform reasoning on these out-of-distribution abstract puzzles.

- Writing is unclear at points: often the metrics (such as entropy) are presented after the results, which makes the paper hard to follow.

---

> ### Author Rebuttal · Authors · 2024-05-29
>
> We appreciate the feedback and will make below points clearer.
>
> **Limited number of examples**: The manually curated examples are similar to instances in the larger datasets (e.g. Mensa), whose example cannot be presented due to copyright issues. We do not intend to imply that they are special. Rather, we aim to **use these examples as qualitative probes** to showcase the blindspots in depth, and show that the issues are ubiquitous in VLMs, spanning stages of perception, reasoning, and verification. We also provide **quantitative results in Table 1,2**, which showed statistical summary on hundreds of samples, supporting our main findings of blindspots in current VLMs.
>
> **Motivation for abstract puzzles**: While VLMs are mostly trained on naturalistic images, evaluation tasks on foundation models often involve significant distribution shifts or completely novel tasks, and foundational models like VLMs are expected to have some capacity for task generalization and zero-shot capabilities. RPM puzzles challenge VLMs to engage in abstract reasoning and pattern recognition, serving as a controlled test for perception, self-reflection, and deductive reasoning abilities. Additionally, IQ evaluations for humans already use abstract visual puzzles, establishing a comparative connection to human performance.
>
> To investigate whether the VLMs' struggle is due to a lack of general reasoning or data distributional shifts, we plan to evaluate VLMs on pattern-based reasoning tasks with naturalistic images, and investigate the impact of additional instruction fine-tuning on RPMs.
>
> **Models without interleaved visuals**: Our submission tested only models capable of handling interleaved inputs. Early stage VLMs sometimes only support image as input. However, it is rare to find SOTA VLMs that cannot handle interleaved visual inputs.
>
> **Entropy calculation**: The entropy is calculated from multiple samples obtained by running the model with different random seeds. p_i refers to the predicted probability distribution over answer choices for a given sample.
>
> **Self-consistency entropy**: We bootstrapped 1000 leave-one-out repetitions from 5 answers per question and calculated entropy over aggregated predictions from each repetition.
>
> **Varying difficulty**: Figure 2 exhibits varying degrees of difficulty based on pattern complexity, number of elements, and underlying reasoning logic required. We will provide a clearer explanation of factors contributing to difficulty levels.

---

> > ### Comment · Reviewer_Hoxc · 2024-06-04
> > **Response to Authors**
> >
> > **Limited number of examples:** While I understand that the qualitative probes are meant to showcase the specific blindspots in VLMs, the results are hardly compelling if the entire analysis of "what are the blindspots" is only based on these 3 examples. These results would be a lot more compelling if the trends were exhibited on a larger set of images (even 10 examples per difficulty level). For example, in Table 4, you show that getting descriptions of segmented images results in fewer hallucinations. However, this finding is based on only one example (M-Medium). If this trend was shown to hold consistently on even ten examples, I would find the results much more convincing.
> >
> > **Motivation for abstract puzzles:** Based on the stated motivation, it seems like you are using visual IQ tests to jointly combine two evaluation axes (image distribution shift and deductive reasoning abilities). However, these two axes are very different to each other, and combining them into a single task does not make for a controlled test at all.
> >
> > I think my main issue with this paper is that it is using an IQ test to evaluate a model, when I don't think this is an appropriate use case for these tests -- IQ tests were designed for humans, not VLMs, and they tell a very different story for VLMs than they do for humans.

---

> > ### Author Response · Authors · 2024-06-06
> >
> > Thank you for the additional comment. We address them in below.
> >
> > **Limited Number of Examples**
> >
> > Our intention was to highlight specific instances that reveal the models’ deficiencies in handling complex visual deductive reasoning tasks. Table 4 only serves as an illustrative case study, presenting three qualitative examples for demonstrating the blindspot. We agree that a broader set of examples would provide a more robust analysis, we have incorporated new results from the Raven subset (60 examples) to expand our figure 3, which shows the same conclusion that additional segmentation dramatically improve the perception of the VLMs:
> > | | Raven | Manual-easy | Manual-medium | Manual-hard |
> > |-----------|-------|-------------|---------------|-------------|
> > | Before Segmentation | 7.4 | 50 | 20 | 20 |
> > | After Segmentation | 21.2 | 80 | 60 | 30 |
> >
> > We also tested many examples in the Mensa and IT datasets, which shows the same finding as well. We will add the results and additional analysis into our paper.
> >
> > **Motivation for Abstract Visual Puzzles**
> >
> > Your point about the decoupling of image distribution shift and deductive reasoning abilities is well taken. As we mentioned in our last comment, we would investigate these pattern-based reasoning tasks using naturalistic images. This will help us assess their performance with realistic objects which does not have data distribution shift. We will also investigate if instruction fine-tuning can help close the data distribution gap and enhance the VLM performance. These studies will help us gain more insights in whether data distribution shift is the culprit. However, they require more effort, making them beyond the scope of this paper, which focuses on identifying existing issues rather than meticulously tracing their origins.
> >
> > Our goal was to simulate a more holistic challenge that mirrors scenarios where generalist VLMs must navigate through even unseen and unfamiliar scenarios. You questioned whether it is appropriate to use RPMs for VLM evaluation. Although the PRMs were designed for humans as what you pointed out, we think that RPM problems represent a fundamental type of visual-spatial reasoning that artificial intelligence systems, particularly those aimed at achieving general intelligence, should also be able to perform well on.
> >
> > In fact, using RPMs to assess VLM is not rare. For instance, many general VLM benchmarks like MATHVISTA [1] and MMMU [2] uses RPMs problems to probe models’ reasoning and understanding ability of complex patterns. When GPT-4v came out, technical evaluation work like [3] also probed its performance on abstract patterns, especially the RPMs. Different from them, our work dives deep into scrutinizing the blindspot, and the underlying issues when VLMs deals with these abstract patterns, showing some potential directions for future improvement.
> > ```
> > [1] Lu, Pan, et al. “Mathvista: Evaluating mathematical reasoning of foundation models in visual contexts.” ICLR 2024.
> > [2] Yue, Xiang, et al. “Mmmu: A massive multi-discipline multimodal understanding and reasoning benchmark for expert agi.” CVPR, 2024.
> > [3] Yang, Zhengyuan, et al. “The dawn of lmms: Preliminary explorations with gpt-4v (ision).” arXiv preprint arXiv:2309.17421 9.1 (2023): 1.
> > ```

---

> ### Comment · Reviewer_Hoxc · 2024-06-07
> **Response to Authors**
>
> Thank you for the additional results. What does the first column ("Raven") mean? is that also an accuracy score?
>
> I do not personally like the use of RPM-style intelligence tests for evaluating VLM reasoning abilities (including in datasets like MathVista and MMMU). I'll quote [Melanie Mitchell](https://aiguide.substack.com/p/did-chatgpt-really-pass-graduate) here about what these intelligence benchmarks like MMLU tell us about ChatGPT:
> > But does an AI system’s performance on an exam actually predict that it will exhibit skills in the real world?  Perhaps there is a correlation between how humans perform on tests and on their future skills in the real world, but that correlation has not been demonstrated for AI systems.
>
> > For example, when a human succeeds in answering a test question such as the example on inventory given above, we assume that the human can then generalize this understanding to similar situations—the point of the test, after all, is to evaluate knowledge and skills that go beyond the wording of any specific question.  But is the same true for ChatGPT?
>
> At the same time, I acknowledge your point that benchmarks like MMMU and MathVista do exist and are generally accepted in the community, so it would be unfair to penalize you too harshly for using similar RPM tests. I will raise my score to a 5, since my other concerns about the paper (especially that most of the qualitative analysis is only done on 3 examples) still stand.

---

> > ### Author Response · Authors · 2024-06-07
> >
> > Thanks for your comment!
> >
> > > What does the first column ("Raven") mean? is that also an accuracy score?
> >
> > We have three datasets in our paper: Raven, IntelligenceTest, and Mensa.
> > The first column in this table is the accuracy of 60 examples from Raven dataset.

---

### Decision · Program_Chairs · 2024-07-10

**Decision:**

Accept

**Comment:**

The authors perform a quantitative and qualitative evaluation of pretrained vision and language models on visual reasoning tasks like Raven, which involve understanding abstract visual patterns. We believe the paper presents novel insights through a well-designed experimental setup and in-depth analysis, making it a valuable addition to the COLM program. The main concern raised during the review process is whether evaluating on abstract puzzles is a convincing testbed for visual deductive reasoning in VLMs, especially since these models are mostly trained on natural images. Testing whether VLMs can generalize to these abstract puzzles is a novel and interesting challenge. However, I encourage the authors to discuss the distribution shift issue in-depth and potential limitations in the final version. Adding results of realistic objects or investigating instruction fine-tuning can certainly strengthen the paper as well. The authors should also discuss the copyright issue in the final version.

[At least one review was discounted during the decision process due to quality]